# Experiences and care trajectories of persons living with Hepatitis B in Senegal: A qualitative study

Mariama Diédhiou[1,2‡], Albert Gautier Ndione[3‡], Judicaël Tine[4], Amady Ndiaye[4], Oumy Camara[4], Kiné Ndiaye[5], Hélène Kane[6], Louise Fortes Deguenonvo[7], Ndeye Fatou Ngom[5], Gilles Wandeler[4,8,9], Moussa Seydi[4], Adrià Ramírez Mena[4,8,10]*, for SEN-B[¶]

1 Centre Régional de Recherche et de Formation à la prise en charge clinique de Fann (CRCF), Fann University Hospital, Dakar, Sénégal, 2 Département de Sociologie, UFR Sciences Économiques et Sociales, Université Assane Seck, Ziguinchor, Senegal, 3 Département de sociologie, Université Cheikh Anta Diop, Dakar, Senegal, 4 Infectious and Tropical Diseases Service, Fann University Hospital, Dakar, Senegal, 5 Centre de Traitement Ambulatoire, Fann University Hospital, Dakar, Senegal, 6 Dysolab, Sociology department, Rouen Normandie University, Rouen, France, 7 Department of Infectious Diseases, Dalal Jamm Hospital, Guediawaye, Senegal, 8 Department of Infectious Diseases, Bern University Hospital, University of Bern, Switzerland, 9 Institute of Social and Preventive Medicine, University of Bern, Bern, Switzerland, 10 Graduate School of Health Sciences, University of Bern, Bern, Switzerland

‡ These authors contributed equally to this work.
¶ Membership of SEN-B is provided in the Acknowledgments.
* adria.ramirezmena@students.unibe.ch

## Abstract

Hepatitis B virus (HBV) infection is a public health concern in Senegal, where individuals encounter multifaceted barriers in their care trajectories. This qualitative study aimed to identify the individual, social, and structural determinants influencing HBV care delivery to inform targeted interventions for improving access. Between February and April 2021, we included 29 adults living with HBV from SEN-B, a prospective HBV cohort in urban Senegal providing biannual clinical monitoring for HBV-related liver complications. We conducted individual semi-structured interviews, during which participants shared their experiences about care trajectories. Through inductive analysis, we identified four primary factors that shape care trajectories: diagnostic circumstances, disclosure choices, social environment, and financial burden. Screening for HBV frequently occurred incidentally during blood donations, illness episodes, or prenatal care with inadequate post diagnosis communication adversely affecting disclosure decisions. While many participants disclosed their status to immediate family, some withheld information due to misconceptions about transmission risks and disease progression. The social environment played a dual role by favoring individuals' motivation to seek advice while also providing conflicting information, particularly regarding healthy and dietary practices. Financial constraints emerged as a critical barrier, limiting access to care and resulting in follow-up interruptions. Participants experienced significant delays between diagnosis and care engagement, often

**Data availability statement:** The data in this study does indeed contain potentially identifiable information about participants (detailed demographic data, precise geographical location). These restrictions were imposed by the National Ethics Committee for Health Research No. 00000061 MSAS/CNERS/CP, which oversees this project. Requests for access to anonymised data can be sent to: Name of data access manager: Babacar Thiendella Faye Institution: Service des Maladies Infectieuses et Tropicales du Centre Hospitalier National Universitaire de Fann Email: thiendella.faye@smit.sn.

**Funding:** This work was supported by the Schweizerischer Nationalfonds zur Förderung der Wissenschaftlichen Forschung (PP00P3_211025 to GW) and the Swiss Cancer Research Foundation (KLS-4879-08-2019 to GW).

**Competing interests:** The authors have declared that no competing interests exist.

exacerbated by financial constraints or periods of being asymptomatic. The research highlights the complex care trajectories shaped by screening contexts, information quality, treatment availability, and financial barriers in Senegal. To mitigate these issues, it is crucial to raise public awareness, streamline care pathways, and establish effective monitoring systems. A coordinated, multi-sectoral approach involving all stakeholders is necessary to improve healthcare access and equity.

## Introduction

In sub-Saharan Africa, nearly 10% of the population is living with hepatitis B virus, with both vertical and horizontal transmission contributing significantly to the ongoing epidemic [1,2]. This high prevalence exposes the region to an increased risk of cirrhosis and liver cancer, both serious complications of chronic infection [3]. A significant proportion of individuals remain asymptomatic for extended periods, leading to delays in diagnosis and subsequent treatment. Despite the goal set by the international community to eliminate hepatitis by 2030, progress in sub-Saharan Africa has been slow and, in many areas, insufficient. According to WHO data, less than 2% of people in the region have received a diagnosis and only 1% of those eligible for antiviral treatment, have started [2,4]. Current care practices have emphasized not only antiviral therapy for people with severe forms of HBV, but also the importance of regular monitoring to assess disease progression. Access to nucleos(t)ide analogs has been crucial for effective disease management. However, these challenges may be exacerbated by economic and structural constraints, including inadequate healthcare resources, a lack of specific national guidelines for HBV management and the concentration of specialised services in urban areas [5–7].

Beyond economic and structural barriers, HBV management in sub-Saharan Africa faces significant sociocultural challenges, including limited disease awareness among general population and healthcare providers, along with persistent stigma [8–10]. Effective management of HBV requires a holistic approach that integrates not only medical care but also the social, economic, and cultural determinants influencing healthcare access [11,12]. In Senegal, economic constraints and limited access to standards of care often compel individuals with chronic infection to seek treatment only during severe illness or exacerbations [13,14]. Although effective treatments and preventive measures, like HBV birth-dose vaccination, are available, many adults face significant barriers in accessing and navigating the healthcare system [15,16].

To improve access to healthcare and enhance health outcomes, it is essential to examine care pathways with a focus on individual experiences. Current research on HBV management remains fragmented. With studies addressing isolated aspects of the disease but lacking a comprehensive analysis of the full individual journey from screening to long-term follow up. Furthermore, while individual perspectives are critical for shaping effective health policies and services, the lived experiences of individuals with HBV remain underrepresented in existing literature [17]. We aimed to address existing gaps in the understanding of care trajectories for individuals

living with HBV in urban Senegal. We used qualitative methods to investigate the individual, social, and structural factors influencing HBV care pathways in participants of SEN-B, a prospective cohort of persons living with HBV in Dakar, The findings of this research are expected to contribute to the formulation of targeted interventions aimed at improving HBV care and health outcomes in Senegal.

## Materials and methods

### Study design and setting

We conducted face-to-face interviews to gain in-depth insights into the experiences, perceptions, and care trajectories of adults living with HBV in an urban area of Senegal. Semi-structured interviews were chosen to allow for flexibility in responses, enabling participants to express their thoughts and feelings in their own words while also ensuring that key topics of interest were covered systematically. The research was conducted at two referral healthcare facilities in Dakar: the Service des Maladies Infectieuses et Tropicales de Fann (SMIT) and the Centre de Traitement Ambulatoire (CTA) at the Fann University Hospital of Dakar, both of which serve as referral clinics for HIV and other infectious diseases. These settings were selected due to their established expertise in managing chronic infectious diseases, as well as the demographic diversity of the populations they serve, offering valuable context for understanding the complexities of living with HBV.

### Study population and sampling

Participants were drawn from the Senegalese Hepatitis B cohort (SEN-B), a prospective cohort study established at the SMIT to monitor long-term liver-related complications of chronic HBV in urban Senegal. SEN-B provides biannual clinical and virological monitoring enabling longitudinal assessment of disease progression [18]. For this study, we recruited a purposively selected sub sample of SEN-B participants enrolled between September 2019 and October 2020. Inclusion criteria comprised adults aged 18 or older with a laboratory-confirmed HBV diagnosis who provided informed consent for participation in qualitative interviews. To ensure heterogeneity in socio-demographic and clinical characteristics, we employed stratified purposive sampling, selecting participants based on age, sex, marital status, time since HBV diagnosis, level of education, and place of residence. We excluded individuals with HIV co-infection, ensuring that our findings would specifically reflect the experiences related to HBV alone.

### Data collection

We used semi-structured interviews for data collection, conducted by two trained social anthropologists (MD and AGN) experienced in qualitative research conducted the interviews. An interview guide based on relevant literature and aligned with the study objectives,featured open-ended questions to elicit detailed responses from participants.The guide examined care trajectories across individual factors (personal beliefs and coping strategies), interpersonal factors (family support and interactions with healthcare providers), community factors (access to resources and social support networks), and societal factors (healthcare policies and stigma).

Before the main data collection phase, a pilot phase was from February 3–15, 2020, involving nine individuals to evaluate the clarity and effectiveness of the interview guide. Feedback from the pilot participants led to refinements in the guide, ensuring it effectively captured relevant aspects of participants' care experiences and trajectories. Subsequently, between April 14 April and June 10, 2021, interviews were conducted with 20 additional participants. Appointments were scheduled in advance by telephone, to accommodate participants' availability. Prior to participation, all individuals received a study overview and provided verbal informed consent. Verbal consent was recorded prior to each interview. Participants' were first asked to share their concerns about the disease and its treatment. The participants' concerns were incorporated into our disease and grounded in real-life experiences, in line with anthropological and person-centred care approaches.

Following each interview, participants received individualized counseling from a study clinician to address their questions, with additional information sessions provided as needed. The face-to-face interviews were conducted in either Wolof or French, lasting between 45–90 minutes, and were held in a designated room at the Fann University Hospital of Dakar, Senegal.

## Data analysis

The interviews were conducted in person and audio-recorded to accurately capture participants' responses. The audio recordings were translated then transcribed Wolof to French by research assistants, CB and BRT. Transcripts were anonymized by assigning pseudonyms to participants to ensure confidentiality. The data collection process continued until data saturation was achieved. The number of interviews required to reach saturation was determined iteratively throughout the study, to ensure the completeness of the data collected accompanied by regular data analysis [19]. We employed inductive approaches, with codes were initially defined based on the study's themes, and new emerging themes identified during the analysis of the interview transcripts. The verbatim transcriptions were then classified by theme. Thematic analysis facilitated the systematic synthesis of the respondents' discourse [20]. Subsequently, a coding report was generated for further analysis.

For the mapping of care trajectories, two researchers (AGN and ARM) reviewed the interview notes to construct a pathway diagram for each participant, illustrating their progression through healthcare resources from initial screening to enrollment in SEN-B. The resources used at each time point were categorized inductively by one researcher (ARM) and subsequently reviewed by two additional researchers (MD and AGN) to ensure consistency in the classification process. The timing of each event was recorded and rounded to the nearest semester (e.g., first or second half of the year). A visual diagram was then developed to map each individual's trajectory through the healthcare system, displaying interactions at the semester level.

## Ethical considerations

Ethical approval for the study was obtained from the National Ethics Committee for Health Research at the Ministry of Health and Social Action of Senegal (0061/MSAS/DPRS/CNERS). Written consent was obtained systematically via a signed form at the time of initial inclusion in the SEN-B cohort. It included the aims, procedures, potential benefits, and risks involved. Participants were made aware of their right to withdraw from the study at any time without needing to give a reason. Verbal consent was reaffirmed and recorded before each individual interview. It included a reminder of the specific objectives of the interview, explicit authorisation for audio recording and confirmation of understanding of the right to interrupt. To ensure confidentiality, the recordings were conducted anonymously, with no use of participants' names.

## Inclusivity in global research

Additional information regarding the ethical, cultural, and scientific considerations specific to inclusivity in global research is included in the Supporting Information (S1 Checklist).

## Results

### Study participants characteristics

We interviewed 29 individuals with a median age of 40 years (interquartile range [IQR]: 27–54), of whom 15 (51.7%) were women. Four of the 29 participants reported having health insurance coverage, and 18 (62.1%) were aware of their HBV status at least two years before enrolling in the study. Two participants had never received formal education and two others had received Qur'anic instruction. Most participants were married (70.4%), while a smaller proportion were single

(29.6%). Additionally, 28 out of 29 participants disclosed their HBV status to at least one family member. Other participants' characteristics are summarized in Table 1.

The care trajectories of participants prior to cohort (SEN-B) enrollment are depicted in Fig 1. The figure illustrates the spectrum of health-related resources accessed over time and highlights episodes of sustained care engagement alongside intervals of disengagement. We identified several factors influencing the individuals care trajectories including [1] the circumstances of disease diagnosis discovery, [2] the choices about disclosure, [3] the role of social environment and [4] the financial burden of care.

### Hepatitis B diagnosis

We observed that screening practices were inconsistently implemented, even within medical settings where such assessments are advocated by established screening guidelines and health policies. The specific conditions and environments in which our participants underwent screening significantly influenced their healthcare trajectories. We identified three primary contexts in which individuals received testing: (i) during voluntary blood donation, (ii) through interactions with healthcare professionals without having explicitly sought care for HBV-related symptoms, and (iii) during antenatal visits.

The majority of participants were tested following blood donation, highlighting a critical juncture in their healthcare experience.

"I donated blood at the national blood transfusion center. When I went to get the results, I was informed that I could no longer donate blood because I had the hepatitis B virus. That is how I discovered I had the virus, in 2005". (**Mbathio, ID #17)**

After participating in blood donation campaigns, some individuals reported not retrieving their test results due to distance to the healthcare facility and time constraints.

"I donated blood with my daughter. But she didn't pick up the results because she works as a housekeeper and doesn't have time to go". (**Tida, ID #29)**

As a result, often they only learned their HBV status after subsequent donations at different locations.

The experiences of participants who remained unaware of their HBV status until their second or third donation exemplify missed opportunities for timely diagnosis and subsequent follow-up care.

For women of reproductive age, antenatal consultations emerged as a common context for the unexpected revelation of HBV status.

"During my sixth pregnancy, I was asked to complete medical tests. When I returned with the results, I showed them to my midwife. She was the one who told me I had hepatitis B. After going home, I was so distressed that my blood pressure rose". (**Mariama, ID # 16)**

Moreover, medical and paramedical students reported undergoing routine HBV screening before or during their internships, reflecting the integration of HBV awareness into their educational or professional experiences.

In the absence of pre-test counseling, individuals frequently learned of their positive HBV status upon receiving test results, which were not always communicated consistently. Typically, it was a healthcare professional who informed about the diagnosis, providing varying levels of explanations based on their own understanding and expertise. This moment of revelation often occurred for individuals who had not initially intended to undergo screening. Individuals' reactions to their test results were often shaped by inadequate communication and counseling following the announcement.

**Table 1. Characteristics of the study participants.**

| Id | Name* | Sex | Age | Marital status | Educational level | Employment | Date of HBV diagnosis | Health insurance | Main screening reason | Status disclosure |
|----|-------|-----|-----|----------------|-------------------|------------|----------------------|------------------|----------------------|-------------------|
| 1 | Aby | F | 24 | Single | Tertiary | Student | 2019 | Yes | Blood donation | Yes |
| 2 | Aliou | M | 54 | Married | Secondary | Security officer | 2011 | No | Community screening | No |
| 3 | Anna | F | 52 | Married | None | Housewife | 2019 | No | Clinical suspicion | Yes |
| 4 | Ansou | M | 47 | Married | Tertiary | Educator | 2018 | Yes | Routine check-up | Yes |
| 5 | Antoine | M | 28 | Single | Tertiary | Journalist | 2006 | No | Community screening | Yes |
| 6 | Bachir | M | 23 | Single | Tertiary | Student | 2005 | No | Blood donation | Yes |
| 7 | Coumba | F | 38 | Divorced | None | Salesperson | 2016 | No | Clinical suspicion | Yes |
| 8 | Fatoumata | F | 23 | Single | Tertiary | Student | 2019 | No | Routine check-up | Yes |
| 9 | Habib | M | 61 | Married | Tertiary | Farmer | 2004 | No | Blood donation | Yes |
| 10 | Ibrahima | M | 57 | Married | Primary | None | 2014 | No | Blood donation | Yes |
| 11 | Julie | F | 56 | Married | Primary | Housewife | 2015 | No | Blood donation | Yes |
| 12 | Khady | F | 53 | Married | Secondary | Salesperson | 2020 | No | Clinical suspicion | Yes |
| 13 | Mame Ami | F | 24 | Married | Primary | Housewife | 2018 | No | Antenatal care | Yes |
| 14 | Mami | F | 52 | Widow | Primary | Salesperson | 2011 | Yes | Blood donation | Yes |
| 15 | Mansour | M | 32 | Single | Tertiary | Journalist | 2015 | Yes | Blood donation | Yes |
| 16 | Mariama | F | 36 | Married | None | Farmer | 2017 | No | Antenatal care | Yes |
| 17 | Mbathio | F | 35 | Married | Tertiary | None | 2005 | No | Blood donation | Yes |
| 18 | Modou | M | 40 | Married | Coranic | Jeweler | 2019 | No | Clinical suspicion | Yes |
| 19 | Momar | M | 23 | Single | Tertiary | Student | 2013 | No | Clinical suspicion | Yes |
| 20 | Mounass | F | 53 | Married | Primary | Storekeeper | 2019 | No | Clinical suspicion | Yes |
| 21 | Moustapha | M | 26 | Single | Tertiary | Student | 2014 | No | Blood donation | Yes |
| 22 | Ndoumbé | F | 27 | Married | None | Tailor | 2019 | No | Clinical suspicion | Yes |
| 23 | Ngala | M | 28 | Married | Primary | None | 2005 | No | Community screening | Yes |
| 24 | Nogaye | F | 42 | Married | Secondary | Caterer | 2019 | No | Blood donation | Yes |
| 25 | Oumy | F | 54 | Married | Secondary | Housewife | 2019 | No | Clinical suspicion | Yes |
| 26 | Saliou | M | 25 | Single | Tertiary | Student | 2017 | No | Routine check-up | Yes |
| 27 | Samba | M | 56 | Married | Secondary | Carpenter | 2018 | No | Clinical suspicion | Yes |
| 28 | Sidy | M | 71 | Married | Coranic | Salesperson | 2019 | No | Routine check-up | Yes |
| 29 | Tida | F | 61 | Married | Primary | Housewife | 2019 | No | Blood donation | Yes |

**Footnote:** *Participant's original names were pseudonymised. **Abbreviations:** M: Male; F: Female

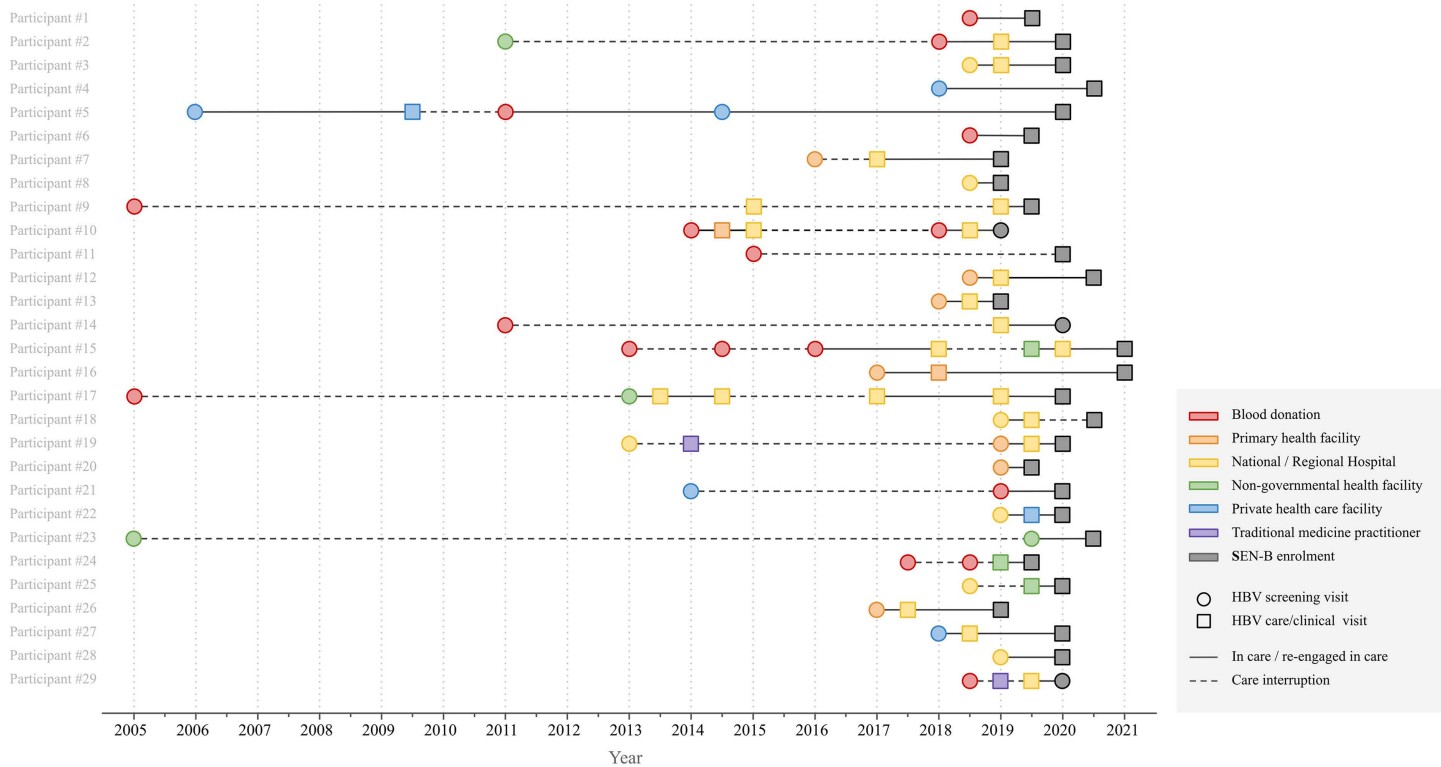

**Fig 1. Diagram of HBV care trajectories by participant (N = 29).**

"I was so discouraged that I got up and left, because he (the doctor) told me that the disease will progress to cancer if I don't get treatment. When I got home, I couldn't stop crying. I kept thinking about this phrase: the disease will progress to cancer." (**Anna, ID #13**)

The way in which information was presented particularly regarding the sexual and hereditary transmission of the disease significantly influenced decisions about whether to disclose their positive status.

## Choices about HBV disclosure

Individuals displayed a spectrum of behaviors when it came to disclosing their HBV status. All but one participant chose to share their status openly. The decision to disclose was influenced by various factors, including the level of proximity and trust in their relationships, the perceptions surrounding the hereditary nature of the disease, and the availability of financial or moral support.

"I have a brother who is currently in France and I talked to him about it. You know when someone gives you their energy and their time, it gives you the courage to talk to them. These people are helping me to pay for my tests, or to pay for transport costs to the hospital. Every time I inform them of my hospital appointments." (**Mami, ID #14**)

The majority of participants opted to disclose their status to specific family members, seeking support and understanding within their intimate social circles. A particular timing pattern emerged concerning the disclosure, which did not necessarily occur immediately following the announcement. Often, individuals choose to reveal their status triggered by tragic

incidents involving others, such as the death of a family member due to HBV-related complications. However, one participant chose not to disclose, mainly due to a lack of knowledge about the disease and the absence of clinical symptoms.

"I didn't tell my wife or my children because I don't think it's as serious as all that to tell them."(**Aliou, ID #2)**

A significant barrier to disclosure was the insufficient information regarding transmission risks, clinical manifestations, the progression of the disease.

This ignorance fuelled self-stigmatisation, often resulting in social self-exclusion. Some participants have adopted excessive behaviours for fear of infecting their loved ones. As the following account shows

"I don't go anywhere anymore, I isolate myself [...]. I also have my own spoon because. I don't want to risk infecting my family." (**Mame Ami, ID #13)**

This quote illustrates how the fear of transmitting HBV, although medically unfounded, can lead to isolation strategies and excessive precautions. This fear and uncertainty frequently resulted in hesitance to share their status, which not only affected the dynamics of family screening practices but also contributed to the trivialization or denial of the illness.

"My children haven't been tested because I don't know what hepatitis is, what its symptoms are. I hear about hepatitis B, but I don't know what it is. If I knew the consequences of the disease, I'd ask them to go." (**Anna, ID #3)**

The analysis revealed two distinct information management strategies: participants with a supportive social network generally shared their status and accessed care more easily through family advice. Others, particularly those without relatives in the healthcare field, developed informational autonomy through online media.

"After the doctor told me I had hepatitis B, I spent the whole day doing research on the internet. I searched on Youtube what hepatitis B is, how it manifests itself, how you get it... I looked at everything. I also heard that traditional medicine can treat the disease and that if the virus hasn't yet reached the liver, it can be used to treat it." (**Nogaye, ID #24)**

The time lapse between diagnosis and linkage to care varied widely, ranging from months to several years (Fig 1). During this time, individuals frequently navigated between various formal and informal healthcare settings, which did not always result in consistent and regular follow-up. Some participants sought medical care and testing through both public and private institutions, while others turned to traditional medicine or practitioners.

"Both of my maternal uncles had hepatitis B. They went to the hospital, but there was no improvement, so they went back to traditional medicines, and that's what they used until they died." (**Fatoumata, ID #8)**

### Role of social and community networks in therapeutic choices

Participants frequently embarked on an individual quest for information regarding potential remedies, particularly focusing on dietary advice about foods that could be beneficial or harmful. The participants appeared to have internalized the belief that reducing fat intake was vital for their health, influenced by a complex network of advice from medical, family or community circles. Their initial therapeutic choices typically focused on food, particularly fruits, such as lemon, papaya, and avocado, widely recognized for their health benefits.

"I heard one of our parents saying that avocado is good for hepatitis B because it cleans the blood. I decided to try it out and bought a kilogram for 1500 FCFA ($2.45) at the market. I started eating one avocado a day and would buy more whenever I had money." (**Tida, ID #29)**

Conversely, we observed individuals cautioning against the consumption of some products, particularly peanut-based products or oilseeds, reflecting the culturally informed link between some nutrients and the risk of malignancies.

"'I've heard that peanuts contain a toxic acid, I can't remember what, but it's

something toxic. You shouldn't eat them." (**Fatoumata, ID #8)**

"At the blood transfusion center, they asked me to stop eating fatty foods like meat fat and peanut paste, which I really liked. So I reduced my intake. They explained to me that fats go straight to the liver, and when it's weakened, it becomes more vulnerable, making it easier for the virus to take hold." (**Nogaye, ID #24)**

Other remedies included medicinal plants, many of which were referred to as "old Senegalese remedies". These plants are esteemed for their liver-protective qualities and are often employed not only for the treatment of liver-related ailments but also for a variety of other health conditions.

"There are traditional medicines that I used and yet I didn't see any harmful side effects for me like "laydour" (*cassia senna*), "nguer" (*guiera senegalensis*), and soursop leaves (...) There is also a medicine called "Forever" that I used. My little brother sells it. It is used to clean the blood (...) [I take] just one glass a day." (**Modou, ID #18)**

Individuals were frequently resorted to these medicinal plants, typically consuming them as infusions or in powdered form. The use of herbal remedies often took place early in the therapeutic process, immediately following the individual's diagnosis of infection. However, there were instances where this reliance on herbal treatments occurred later, particularly after individuals have engaged with conventional medical consultations that did not meet their expectations. While older generations generally exhibited a greater openness to traditional medicine, younger individuals frequently conveyed skepticism or outright rejection of these practices, particularly when it concerned the health and treatment of their children.

"I didn't use traditional medicines. I didn't use them for myself, and I didn't give them to my daughter either. This was because I was unaware of the ingredients or contents of these medicines, and I was not sure about how they would affect my liver." (**Mbathio, ID #17)**

Alongside the insights garnered from individuals' immediate social environment, including friends and family, the Internet served as an invaluable resource for information regarding therapeutic products and healers.

"I am part of a Facebook group where a woman anonymously shared that she has the disease. She mentioned she has been undergoing treatment with traditional medicine but hasn't noticed any improvement. She requested recommendations for a reliable traditional practitioner. Several group members suggested professionals she could contact." (**Ndoumbé, ID #29)**

Some individuals sought information about HBV through online and social media platforms, navigating a landscape where traditional healers actively promote homemade herbal remedies and assert their capacity to cure the disease. Although biomedical information was accessible, the prominence of traditional healers providing these treatments rendered them a favored option for many.

**Beyond financial expenses**

Participants' care trajectories often involved navigating multiple healthcare facilities, frequently guided by knowledgeable relatives who directed them to appropriate medical services during symptomatic phases. These transitions between

facilities included receiving referrals for further testing or connecting with the local hepatitis B patients' association, Saafara Hépatite Sénégal, which facilitated timely and reliable access to appropriate care.

> "I disclosed my status with my little sister who works at Fann Hospital. I can say that it was thanks to her that I got an appointment quite quickly." (**Khady, ID #12**)

> "I was referred to the Fann Hospital by the president of the HBV patients association (Saafara Hépatite Senegal)." (**Mansour, ID #15**)

Participants with relatives working in the health sector often have relied on modern medicine as their primary resource for care. However, financial constraints frequently hindered their ability to afford multiple follow-up tests and examinations, resulting in interruptions of care, with the financial burden commonly cited as a key reason for discontinuation.

> "I stopped follow-up when my father retired because consultations cost 15,000 CFA (25 US$) and he could no longer afford to pay them. And I stayed like that until 2014 when I started having physical problems and I went to visit the other doctor." (**Moustapha, ID #21**)

> "As I said, it's the high cost of the tests: when the cheapest one is 14,000 FCFA (23 US$) and you have to do nine different tests for a general check-up, it's difficult. Once I was asked to do a full blood check-up including 16 or 17 different tests: I used to do groups of 4 or 5 at a time until I did them all, because it was binding and financially very cumbersome." (**Momar, ID #19**)

The financial burden of ongoing HBV care compelled not only individuals but also their support networks (family and friends) to mobilise resources. The cost of frequent consultations and tests often led participants to discontinue their follow-up care, as they had exhausted their own resources and experienced the weight of continuously relying on their social circle.

> "I was actually so used to going to hospital that I was told I had a house there, so much so that I stopped going there for my visits". (**Mami, ID #14**)

In the case of Mami, the financial strain experienced by her family and friends manifested as a form of "resident" stigma within the hospital environment, ultimately forcing her to discontinue her follow-up care. Moreover, beyond financial considerations, various factors related to the medical information received from healthcare practitioners, also contributed to the decision to halt medical follow-up.

> "I was diagnosed at Pikine hospital. I was prescribed further tests to be completed at Dantec Hospital. Once I'd received the results, I took them back to Pikine, where the doctor asked me to go to Fann Hospital to complete the assessment. But as the other doctor told me (Hepatitis B) was an incurable disease, I left all the papers and visits aside." (**Modou, ID #18**)

Participants were confronted with an overwhelming and often inconsistent flow of information throughout the diagnostic process. The situation was exacerbated by the fragmentation of referrals between health facilities and the financial burdens associated with continuing care, which often led people to discontinue their treatment.

> "I was followed by a gastroenterologist, who prescribed a series of tests every six months. I completed the tests and reported the results. The results remained the same, and the tests were expensive. After three years of follow-up, I stopped going because I didn't want to burden my parents with any more expenses for a disease that has no cure." (**Antoine, ID #5**)

The multiplicity of needs (food, housing, healthcare costs) and the instability of income led some individuals to adopt survival strategies that could include reducing or abandoning medical care, reflecting an economic rationality constrained by precariousness. While financial constraints pushed some people to discontinue care, others stopped care out of frustration at the stagnation of disease improvement despite the absence of symptoms, or as a result of repeated consultations with healthcare providers who offered little in the way of solutions, clarity or hope.

"I went to the hospital in Grand Yoff, where the doctor ordered blood tests and an abdominal ultrasound. After completing the tests, I returned with the results. The doctor informed me that the results were fine and decided to repeat the same tests during my second visit. During the third consultation, I repeated the same steps, but the doctor mentioned that there were only "traces" and reassured me that it wasn't a serious issue. That's why I stopped [follow-up]." (**Mansour, ID #15**)

The information received when the disease was announced strongly influenced participants' decisions about whether or not to start medical follow-up. Expressions such as "traces of hepatitis" or "incurable disease" often discouraged some participants from seeking care. The choice of using health structures, private practitioners, or traditional healers shaped or interrupted their care trajectories. Those who initially consulted healers or concealed their status due to stigma often encountered obstacles that delayed their access to needed medical care.

## Discussion

In an environment characterized by widespread ignorance about Hepatitis B virus (HBV), the choices surrounding care and treatment for the Senegalese population living with HBV were influenced by a multitude of interrelated factors. The circumstances of testing often lacked adequate preparation and support, while the quality of information provided by healthcare practitioners significantly impacted individuals' understanding of their health condition. Poor communication regarding positive test results led to confusion and anxiety, complicating healthcare decisions. Additionally, the influence of relatives and social networks played a critical role, as discussions about health issues were mediated by familial and community perceptions that could either encourage or deter individuals from seeking necessary medical care. Notably, the discovery of one's HBV-positive status often occurred involuntarily rather than through proactive testing, resulting in a disjointed response where diagnosis did not consistently lead to the pursuit of medical care. The interplay of awareness, information quality, social influences, financial constraints and the context of diagnosis highlighted the complex social determinants that shaped the healthcare trajectories of those living with HBV in Senegal, indicating a need for targeted interventions to address these issues.

Access to and uptake of HBV testing in Senegal, as in many African countries, remains limited [21,22]. Our research found that HBV-positive status is often discovered incidentally during blood donations, illness episodes, or routine prenatal examinations. These findings are consistent with studies from other West African settings, which reported that HBV testing is often primarily targeted towards specific populations, such as blood donors, symptomatic individuals or people living with HIV [17,23,24].Notably, the majority of participants in our study reported first receiving information about HBV only after being diagnosed, underscoring substantial gaps in population-level awareness and proactive screening initiatives. In contrast, higher levels of knowledge about HBV have been described among pregnant women, likely due to proactive information dissemination in antenatal healthcare settings [25]. Studies investigating barriers to HBV testing across African facilities have consistently reported low levels of HBV knowledge among healthcare professionals [26–29]. These deficiencies extend more broadly to HBV awareness, as confirmed by studies conducted in various contexts, affecting both healthcare providers and the general population [30,31]. Collectively, these findings indicate a pervasive and generalized lack of understanding of HBV representing a significant barrier to accessing testing services. This observation is consistent with the findings from low-endemic countries such as the US, where similar gaps in understanding of the modes of

transmission have been documented [32]. Participants in our study reported that the information provided during the initial announcement of their HBV diagnosis was often inadequate and difficult to understand. The perceived technicality of status announcements generated confusion, eroding trust in healthcare providers and prolonging care-seeking delays. This contributed to protracted intervals between diagnosis disclosure and formal healthcare engagement, exemplifying patterns of medical wandering—a phenomenon previously documented in sub-Saharan Africa [33]. Such trajectories were shaped by intersecting structural vulnerabilities, psychosocial barriers, and cultural interpretations of illness, underlining the need for contextually adapted chronic care models.

Our participants' prolonged medical journeys highlight the critical need for effective communication and support from the moment individuals receive their diagnosis. Similar findings have been reported in West African contexts, where unclear and unstructured information, along with inadequate post-diagnosis counseling, triggered strong emotional responses and diminished individuals' willingness to seek care [5,23,34,35]. Effective health communication is a critical determinant of disease management, particularly for chronic conditions such as diabetes and HBV. Inadequate communication can lead to significant gaps in an individual's understanding of disease mechanisms, adversely affecting adherence and clinical outcomes. For instance, a study conducted in Mali demonstrated that limited communication about diabetes often resulted in poor disease comprehension [36], which in turn fostered feelings of helplessness and contributed to care interruptions or discontinuation.

Upon receiving their positive HBV status, healthcare professionals encouraged individuals to inform their partners and family members to promote screening among close relatives. Most participants chose to disclose their status, driven by the need for psychological support and financial assistance. This openness frequently resulted in familial screening, highlighting the importance of collective care, as noted by Coutherut and Desclaux (2012) in the context of HIV [37]. However, reluctance to disclose HBV status often stemmed from misconceptions about transmission and fears of stigma and rejection. Our study found significant self-stigmatization among participants, consistent with existing literature that identifies stigma as a major barrier to disclosure and access to care [38]. Similar studies conducted both in HBV endemic and non-endemic regions have also documented stigmatizing attitudes, reflecting social avoidance, and thus contributing to social isolation and delayed access to care [39–41]. Similar studies from West Africa further emphasized the pivotal role of stigma in shaping care pathways of individuals living with HBV [23,35,42]. This fear of rejection and misconceptions has created missed opportunities for community and household screening, which is vital for improving testing rates and achieving elimination goals. Additionally, the asymptomatic nature of HBV has often resulted in an underestimation of its severity, contributing to delays in seeking care, as noted by Adjei et al. (2019) in Ghana [35].

The social environment played a decisive role in care-seeking behaviors among participants, with some resorting to traditional medicine. Senegal is characterized by therapeutic pluralism in the management of chronic diseases, including HBV, reflecting deeply embedded sociocultural practices [43]. This pluralism has often resulted in fragmented care trajectories, where individuals navigate between testing and biomedical treatment [12]. Traditional beliefs in Senegal and many other sub-Saharan African settings promote the use of medicinal plants and our data indicated that perceptions of efficacy and financial accessibility drive the choice to pursue traditional treatments for HBV [44,45]. Our study, reflecting those of Mugisha et al. (2019) and Nsibirwa et al. (2020), confirmed a pattern of using medicinal plants in the treatment of hepatitis [46,47]. This recourse to traditional practices was often motivated by inadequate access to biomedical care. The media's coverage of traditional practitioners claiming to treat and cure HBV has intensified public debate about these practices' efficacy [9]. Consistent with previous studies reliance on traditional medicine may delay access to biomedical treatments and increase the risk of complications [5]. Our findings suggested that transitions to biomedical care frequently followed the acquisition of formal knowledge about the disease or the death of a close relative from HBV. Nevertheless, fears, fragmented information, and healthcare costs could prolong this transition, leading to significant delays in receiving appropriate care.

The therapeutic trajectories observed in our study have illustrated a multifactorial phenomenon prevalent in health systems where modern and traditional practices coexist. Participants engaged in diverse care practices influenced by socioeconomic constraints, social networks, and prevailing disease representations. Financial inaccessibility often hindered follow-up and commitment to care, resulting in erratic care pathways that undermine the WHO's hepatitis elimination goal of treating over 80% of those in need. A recent study in The Gambia highlighted significant gaps in the HBV care cascade, revealing that only 33% of individuals who sought healthcare received a complete initial liver assessment, despite 70% seeking care [48]. As shown in previous studies, high costs of screening and biannual follow-up examinations, coupled with a lack of referral laboratories at decentralized healthcare facilities, created barriers to prevention and access to timely and appropriate care [8,28]. These financial barriers have contributed to the use of multiple therapeutic approaches, leading to fragmented care trajectories. This situation emphasized the complex interplay between economic factors and healthcare access, underscoring the urgent need for a more integrated and equitable approach to HBV care that addresses both financial and systemic disparities.

Our study is among the first to highlight the diverse sequences of experiences encountered by individuals living with HBV throughout their healthcare journey in West Africa. By analyzing the healthcare-key events in their lives, we identified a wide range of experiences and highlighted major gaps in the existing care offer which underline the urgent need to rethink care strategies to guarantee better care for people living with HBV in the region. Our findings allowed us to improve our understanding of delays in linkage and the reluctance to seek care among affected populations in Senegal, offering valuable insights that could inform national health policies. However, the potential for selection bias may have masked other trajectories, and generalizations from our findings should be approached with caution. Additionally, reliance on participants' experiences may have resulted in the omission of critical information regarding their healthcare journeys. Our methodology did not allow for triangulation with other sources, such as healthcare providers, or individuals not engaged in care, which may limit our ability to fully comprehend the healthcare trajectories within the general population.

## Conclusion

Our study highlighted the fragmented care pathways of individuals with HBV in urban Senegal, revealing critical public health concerns linked to low awareness and uncoordinated testing and linkage to care efforts. Barriers such as accidental diagnoses and stigma around disclosure hindered timely access to care and worsened health disparities. To improve the management of HBV in Senegal, it is essential to implement a comprehensive strategy, including large-scale awareness campaigns to inform the general public about the disease and how to prevent it. At the same time, it is necessary to simplify care pathways by strengthening coordination between the various health care stakeholders. Finally, the implementation of an information system will make it possible to monitor people living with the HBV progress and evaluate the effectiveness of interventions, thus promoting the continuous adaptation of strategies. A multi-sectoral approach, involving all stakeholders, is essential to reduce the impact of HBV in Senegal.

## Supporting information

**S1 Checklist. SEN-B Care Inclusivity PLOS Checklist.**
(DOCX)

**S1 File. Semi-structured interview guide.**
(PDF)

**S2 File. Checklist Standards for Reporting Qualitative Research (SRQR) for qualitative methodology.**
(PDF)

## Acknowledgments

We warmly thank all participants for their involvement in this study. A special acknowledgement for Ibrahima Gueye, Fatou Nguirane and Babacar Coulibaly from SAAFARA Hpatite patient's association in Senegal. We thank all the SEN-B team, in particular Amady Ndiaye, Oumy Camara, Ahmadou Mboup, Aminata Diallo, Khady Ndaw, Khady Ghassama, Sadio Ba, Alassane Ndiaye, Abibatou Diaw, Assietou Gaye, Betty Fall, Alassane Ndiaye, Aminata Ndoye Badji, Melissa Sandrine Pandi, Hubert Messan Akotia, Bineta Seck Fall, Falilou Sow, Mariama Diedhiou, Albert Gautier Ndione, Astou Diop, Aissatou Niang, Boly Niang, Khadim Faye, Fatou Diop, Massaly Diop, Bintou Fall, Marie-Joseph Dieme, Abdoulaye Keita, Bintou Fall, Kevine Tiogouo, Marianne Berthé, Maguette Fall Ndeye, Bruce Wembulua, Babacar Thiendella Faye, Judicaël Tine, Badiane Aboubakar Sidikh, Daye Ka, Kiné Ndiaye, Louise Fortes, Adrià Ramírez Mena, Ndeye Fatou Ngom, Gilles Wandeler, and Moussa Seydi. We extend our gratitude to Gabriele Laborde Balen for her guidance on the manuscript.

## Author contributions

**Conceptualization:** Mariama Diédhiou, Albert Gautier Ndione, Adrià Ramírez Mena.

**Data curation:** Mariama Diédhiou, Judicaël Tine, Amady Ndiaye, Oumy Camara.

**Formal analysis:** Mariama Diédhiou, Albert Gautier Ndione, Adrià Ramírez Mena.

**Funding acquisition:** Gilles Wandeler.

**Investigation:** Mariama Diédhiou.

**Methodology:** Mariama Diédhiou, Albert Gautier Ndione, Adrià Ramírez Mena.

**Resources:** Kiné Ndiaye, Louise Fortes Deguenonvo, Ndeye Fatou Ngom, Gilles Wandeler, Moussa Seydi.

**Software:** Albert Gautier Ndione, Adrià Ramírez Mena.

**Supervision:** Albert Gautier Ndione, Adrià Ramírez Mena.

**Validation:** Albert Gautier Ndione, Adrià Ramírez Mena.

**Visualization:** Adrià Ramírez Mena.

**Writing – original draft:** Mariama Diédhiou, Albert Gautier Ndione.

**Writing – review & editing:** Mariama Diédhiou, Albert Gautier Ndione, Héléne Kane, Louise Fortes Deguenovo, Ndeye Fatou Ngom, Gilles Wandeler, Moussa Seydi, Adrià Ramírez Mena.

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
