## [Decision Letter · Decision Letter 0]

28 May 2025

PGPH-D-25-00055

Experiences and Care Trajectories of Persons Living with Hepatitis B in Senegal : a qualitative study

Dear Dr. Ramírez Mena,

Thank you for submitting your manuscript to PLOS Global Public Health. After careful consideration, we feel that it has merit but does not fully meet PLOS Global Public Health’s publication criteria as it currently stands. Therefore, we invite you to submit a revised version of the manuscript that carefully considers the reviewers' feedback and directly addresses the points raised during the review process.

We look forward to receiving your revised manuscript.

Kind regards,

Edina Amponsah-Dacosta, Ph.D., MPH

Academic Editor

Journal Requirements:

Additional Editor Comments (if provided):

Reviewers' comments:

Reviewer's Responses to Questions

**Comments to the Author**

1. Does this manuscript meet PLOS Global Public Health’s publication criteria?

Reviewer #1: Yes

Reviewer #2: Partly

2. Has the statistical analysis been performed appropriately and rigorously?

Reviewer #1: N/A

Reviewer #2: N/A

3. Have the authors made all data underlying the findings in their manuscript fully available (please refer to the Data Availability Statement at the start of the manuscript PDF file)?

Reviewer #1: Yes

Reviewer #2: Yes

4. Is the manuscript presented in an intelligible fashion and written in standard English?

Reviewer #1: Yes

Reviewer #2: No

Reviewer #1: The manuscript is well written, and its highly relevant given the WHO’s 2030 target for hepatitis elimination and the persistent challenges in HBV care in sub-Saharan Africa. The study contribute on individual lived experience in Senegal. It's well executed with clear relevance to public health

Reviewer #2: Congratulations on this worthy and interesting research, which includes data that has really interesting implications for public health responses to hepatitis B. I would like to make several suggestions to strengthen the manuscript:

• The manuscript is generally well written, although there are several times when grammatical choices needed to be reviewed. I would suggest an independent edit of the manuscript and challenge assumptions made by the authors including where several sentences required a verb or where there was confusion about past and present tense. One minor issue in the abstract was noting that the aim of the manuscript was “identifying individual, social, and structural determinants influencing HBV care pathways thereby informing targeted interventions to improve HBV care delivery”, when the research was focussed on influencing access to care delivery. The use of HBV or hepatitis B needs to be consistent across the manuscript, and hepatitis B is not a heredity disease.

• I note that the journal does not have a word limit; this is one manuscript that would be strengthened by a word count

• There are several times when sentences have been repeated with only a minor revision.

• Sections of the introduction are better included within the discussion.

• I was unable to check statements that had been referenced as I was unwilling to use Zotero.

• I’m not sure framing the analysis of the data along care pathways – which should be fully described for readers – and whether or how this related to the “mapping of care trajectories”. I would suggest that the hepatitis B cascade of care used by the WHO be used as a framing device for the results. This would give the authors the opportunity to reflect a commonly understood logical progression, and where barriers to the different aspects of the lived experience of people with hepatitis B in Senegal such as stigma, the use of traditional medicines and economic burden of health service access could be addressed.

• While recognising that African countries have a specific social, cultural and historical context in responding to hepatitis B, many of the issues raised in the results have been found in other countries, and which could be referenced.

• In relation to the Material and Methods:

o The use of SEN-B to recruit participants is an interesting choice given the intrinsic bias of people with hepatitis B who enrolled into a hepatitis B study who would be more likely to be engaged in the health system and being more likely to have greater knowledge of the condition.

o The interview guide needs to be included

o More information is needed to rationalise why “participants were asked to express their concerns about the disease and its treatment.” and how this affected the research

o How was data saturation determined? The abstract only mentions inductive – how was deductive analysis used?

o The phrase “corresponding semester” needs clarification, as does the provision of consent which is described as both verbal and written.

• Results:

o Does “Date of screening” in Table 1 actually mean date of diagnosis?

o Several quotes are used to illustrate a point but that do not relate to the introduction to the quote or the broader preceding paragraph, while a quote for women being diagnosed through antenatal consults would be useful.

o Assumptions are made throughout the manuscript of knowledge of the Senegal health system. One example is that it is unclear why a person would need to be “retrieving their test results”.

o The following paragraph included in page 13 needs clarification – it sounds really interesting but is very unclear in its meaning: Those who chose to share their positive status … through media sources, particularly online platforms.

o Linkage to care requires a section in and of itself, and the section describing the role of social environment on therapeutic journeys could be more clearly and accurately titled.

o Many of the findings included in the discussion have been found in other settings and could be referred to, particularly in the non-clinical aspects of living with hepatitis B.

o Issues such as “medical wandering” could be better defined, with self-stigmatisation not being widely described in the results, while the relationships between therapeutic pluralism and sociocultural practices is unclear.

This research is important and its findings are necessary for the development of an effective public health response to elimination hepatitis B as a public health threat.

**Do you want your identity to be public for this peer review?** For information about this choice, including consent withdrawal, please see our Privacy Policy

Reviewer #1: **Yes: ** Azwidowi Lukhwareni

Reviewer #2: No

---

## [Decision Letter · Decision Letter 1]

12 Oct 2025

Experiences and Care Trajectories of Persons Living with Hepatitis B in Senegal : a qualitative study

PGPH-D-25-00055R1

Dear Dr. Ramírez Mena,

We are pleased to inform you that your manuscript 'Experiences and Care Trajectories of Persons Living with Hepatitis B in Senegal : a qualitative study' has been provisionally accepted for publication in PLOS Global Public Health.

Best regards,

Edina Amponsah-Dacosta, Ph.D., MPH

Academic Editor

Reviewer Comments:

Reviewer's Responses to Questions

**Comments to the Author**

Reviewer #2: All comments have been addressed

publication criteria?

Reviewer #2: (No Response)

3. Has the statistical analysis been performed appropriately and rigorously?

Reviewer #2: (No Response)

4. Have the authors made all data underlying the findings in their manuscript fully available (please refer to the Data Availability Statement at the start of the manuscript PDF file)?

Reviewer #2: (No Response)

5. Is the manuscript presented in an intelligible fashion and written in standard English?

Reviewer #2: (No Response)

Reviewer #2: Congratulations on the revised manuscript and your thoughtful responses to my previous comments. There is minor editorial support needed, but the paper is a useful addition to viral hepatitis elimination.

**Do you want your identity to be public for this peer review?** For information about this choice, including consent withdrawal, please see our Privacy Policy

Reviewer #2: **Yes: ** Jack Wallace
